# Beyond the Virus: The Collateral Impact of COVID-19 on Antimicrobial Consumption, Microbial Resistance, and Pharmacoeconomics

**DOI:** 10.3390/pathogens14111126

**Published:** 2025-11-05

**Authors:** Alessandra Gomes Chauvin, Isabele Pardo, André Luís F. Cotia, Isabella L. Rosmino, Tatiana A. Marins, Leandro Martins dos Santos, Barbara Barduchi, Alexandra R. Toniolo, Roberta G. dos Santos, Daniel T. Malheiro, Anderson P. Scorsato, Elivane da Silva Victor, Michael B. Edmond, Silvana Maria de Almeida, Alexandre R. Marra

**Affiliations:** 1Hospital Israelita Albert Einstein, São Paulo 05652-900, SP, Brazil; andre.cotia@einstein.br (A.L.F.C.); isabella.rosmino@einstein.br (I.L.R.); tatiana.marins@einstein.br (T.A.M.); leandro.msantos@einstein.br (L.M.d.S.); barbara.barduchi@einstein.br (B.B.); alexandra.toniolo@einstein.br (A.R.T.); roberta.gsantos@einstein.br (R.G.d.S.); daniel.malheiro@einstein.br (D.T.M.); anderson.scorsato@einstein.br (A.P.S.); elivane.victor@einstein.br (E.d.S.V.); silvana.almeida@einstein.br (S.M.d.A.); alexandre-rodriguesmarra@uiowa.edu (A.R.M.); 2Faculdade Israelita de Ciências da Saúde Albert Einstein, Hospital Israelita Albert Einstein, São Paulo 05652-900, SP, Brazil; 3Department of Medicine, School of Medicine, West Virginia University, Morgantown, WV 26506-9111, USA; 4Department of Internal Medicine, University of Iowa Health Care, Iowa City, IA 52242-1007, USA

**Keywords:** COVID-19, antimicrobials, antimicrobial resistance, healthcare-associated infections, pharmacoeconomics, pharmacovigilance

## Abstract

Background: The COVID-19 pandemic had major global repercussions for hospitalized patients, affecting multiple aspects of hospital care. Understanding these effects is important for improving healthcare management and infection control practices. This study aimed to analyze and compare the pandemic’s impact on antimicrobial use in hospitalized patients, with emphasis on therapeutic, microbiological, and pharmacoeconomic aspects. Methods: A retrospective observational study was conducted at a Brazilian tertiary hospital (2018–2022). Adult patients receiving antimicrobials were included. Variables analyzed were antimicrobial consumption, incidence of healthcare-associated infections, resistance profiles, hospital costs, adverse drug reactions, and pharmacy activities. Data were obtained from anonymized institutional records and analyzed using descriptive statistics, time series, and linear regression. Results: Analysis of 268,713 hospitalizations showed that the median monthly number of patients receiving antimicrobials did not increase significantly during the pandemic. Higher consumption of carbapenems, glycopeptides, polymyxins, and echinocandins was linked to more healthcare-associated infections by multidrug-resistant organisms. *Clostridioides difficile* infections declined. Mortality rose significantly, especially among COVID-19 patients. Costs increased by 39%, with antimicrobial-related expenses up 45.7%. Conclusions: The pandemic intensified antimicrobial use, resistance, and costs. While limited by its single-center design and the use of all inpatients as the denominator, these findings highlight the important need for reinforced antimicrobial stewardship to build healthcare system resilience.

## 1. Introduction

Antimicrobial resistance is one of the most serious threats to global public health. It is defined as the acquired ability of microorganisms, such as bacteria, viruses, fungi, and parasites, to survive and multiply despite exposure to drugs that were previously effective. This phenomenon compromises treatment efficacy, complicates infection control, and increases the risk of resistant pathogens [1,2].

Recent studies estimate that resistant infections directly caused over 1.2 million deaths in 2019, surpassing other major infectious diseases such as HIV/AIDS and malaria. Beyond individual health outcomes, antimicrobial resistance generates significant economic and social burdens by prolonging hospital stays, increasing healthcare costs, and raising mortality rates [3].

In response, the World Health Organization (WHO) stresses the need for coordinated and sustainable strategies to contain antimicrobial resistance. Among these, Antimicrobial Stewardship Programs aim to optimize drug use, ensure appropriate prescribing, and reduce the emergence of resistant strains, thereby improving clinical outcomes [4].

However, this challenge was exacerbated by the COVID-19 pandemic, declared by the WHO in March 2020. Early uncertainty in clinical management led to widespread use of antimicrobials, even for viral infections. This practice accelerated resistance, heightened adverse events, increased hospital-acquired infections, and raised healthcare costs. Simultaneously, overburdened health systems revealed structural weaknesses that hindered the implementation of effective infection control measures [5,6].

A recent systematic review found that antibiotics were extensively prescribed to COVID-19 patients despite limited evidence of bacterial coinfection [7]. The review emphasized that antibiotic therapy should be restricted to cases with strong clinical suspicion of bacterial coinfection or superinfection, highlighting a significant gap between prescribing practices and guideline recommendations [7].

In hospitals, particularly intensive care units (ICUs), the combined risks of invasive devices, prolonged hospital stays, and immunosuppression in critically ill patients fostered the emergence of multidrug-resistant microorganisms. This underscores the importance of evaluating how the pandemic influenced antimicrobial use, resistance rates, healthcare costs, and patient safety [8].

This study primarily focuses on antibacterial drug use, as these agents represent the most widely prescribed antimicrobials and the main drivers of resistance during the COVID-19 pandemic. Nevertheless, antifungal and antiviral utilization are discussed where relevant to provide a broader understanding of antimicrobial stewardship practices.

This study aims to assess the impact of COVID-19 on antimicrobial use in hospitalized patients, with emphasis on its association with resistance, nosocomial infections, healthcare-related costs, and pharmacists’ workload. The findings are expected to guide evidence-based antimicrobial stewardship strategies and strengthen post-pandemic health policies.

## 2. Materials and Methods

This observational, retrospective study was conducted at Hospital Israelita Albert Einstein (São Paulo, Brazil), a private tertiary academic medical center recognized for excellence in clinical care, education, and research. The hospital has 950 beds, including 60 in the adult ICU. Hospitalized adult patients (≥18 years old) who received antimicrobials between January 2018 and December 2022 were included. The study period was divided into the “pre-pandemic” phase (January 2018–December 2019) and the “pandemic” phase (January 2020–December 2022).

Antimicrobial consumption was monitored monthly and annually using hospital pharmacy records and electronic systems. Two standardized metrics were applied: defined daily dose (DDD/1000 patient-days), a WHO-recommended measure for international comparisons, and Days of Therapy (DOT/1000 patient-days), considered more suitable for assessing clinical impact. Therapeutic classes analyzed included antibacterials (cephalosporins, carbapenems, glycopeptides, macrolides, and polymyxins), antifungals (triazoles, echinocandins, and polyenes), and antivirals (acyclovir, ganciclovir, oseltamivir, and remdesivir) [9,10].

Infection rates were obtained from the Hospital Infection Control Service. Infections included ventilator-associated pneumonia per 1000 ventilator-days, urinary tract infection associated with a urinary catheter per 1000 catheter-days, bloodstream infection associated with a central venous catheter per 1000 catheter-days, and *Clostridioides difficile* infections per 10,000 patient-days. We distinguished VAP cases occurring in the step-down unit (SDU) from cases occurring in the ICU because in the SDU, pneumonia was associated with non-invasive ventilation in patients with a tracheostomy, whereas in the ICU, it was associated with invasive mechanical ventilation. However, BSI and UTI data were collected from the entire hospital. Isolated microorganisms were classified by resistance profile (resistant, multidrug-resistant, or susceptible). Special attention was given to the ESKAPE pathogens (*Enterococcus faecium*, *Staphylococcus aureus*, *Klebsiella pneumoniae*, *Acinetobacter baumannii*, *Pseudomonas aeruginosa*, and *Enterobacter* spp.) due to their clinical importance [11].

The Health Economics Unit performed the cost analyses, including total hospitalization costs, specific antimicrobial expenditures, and stratification by COVID-19 diagnosis. The results were expressed in Brazilian reals (R$) and compared between periods. Values were converted from Brazil’s currency, the real (R$), to U.S. dollars (US$). The average exchange rate was determined using daily rates published by the Central Bank of Brazil during the last year of the study (2022). On average, USD 1 = BRL 5.16 was used for conversion.

Adverse drug events were recorded by the Medication Safety and Drug Information Service using WHO-standardized terminology. Events were categorized by system organ class, including cutaneous, gastrointestinal, respiratory, renal, hepatic, hematologic, immunologic, neurologic, psychiatric, and musculoskeletal disorders [12]. The term “number of pharmaceutical activities” refers to the total count of all documented professional actions performed by clinical pharmacists. This measure serves as a proxy for total clinical workload volume, representing the sum of all recorded interventions without specifying their individual nature, thus reflecting the overall intensity of pharmacy services.

This study was approved by the Ethics Committee of the Hospital Israelita Albert Einstein (protocol 5.464.155; CAAE: 83487924.6.0000.0071). All the procedures followed the principles of the Declaration of Helsinki (1975, revised 2013). The Ethics Committee waived the requirement for informed consent.

### Statistical Analysis

Graphical analyses and statistical tests were used to evaluate trends, seasonality, and breaks in the time series. Stationarity and consistency were assessed using Cox–Stuart, Fisher, and Dickey–Fuller tests, and autocorrelation analyses. For the construction of tables and figures, quantitative variables were described using mean, standard deviation, minimum, maximum, median, and quartiles. Qualitative variables were described by frequencies. Given the differing lengths of the pre-pandemic and pandemic periods, monthly means and medians were calculated as the appropriate measures for comparing continuous variables such as hospitalizations, deaths, and antimicrobial consumption over time. The analysis relied on aggregated institutional data, and thus, the issue of missing data at the patient level was not applicable.

Data normality was assessed using the Shapiro–Wilk test. Based on this assessment, the comparison of outcome variables between the two periods was performed using Student’s t-test for normally distributed data or the Mann–Whitney U test for non-normally distributed data. Factors associated with antimicrobial consumption were evaluated with linear regression models (ordinary least squares (OLS) and weighted least squares (WLS)). Models examined correlations between hospitalizations, hospital infections, costs, mortality, pharmacy activities, and adverse reactions. Stepwise adjustments were applied, and diagnostics for autocorrelation and heteroscedasticity were performed using Durbin–Watson and Breusch–Pagan tests. A significance level of 5% was adopted, and analyses were conducted in R (version 4.1.1; R Foundation for Statistical Computing, Vienna, Austria) [13,14,15,16,17].

## 3. Results

### 3.1. Hospitalizations and Mortality

During the study period, 268,713 adult hospitalizations were recorded. When adjusted for the differing period lengths by analyzing monthly medians, the median number of hospitalized patients was similar before and during the pandemic (4525 [4266–4706] vs. 4.583 [3897–5273] patients per month, *p* = 0.586). Similarly, the median monthly number of patients receiving antimicrobials showed no significant changes (2334 [2257–2459] vs. 2411 [2121–2620], *p* = 0.399).

The median monthly number of deaths increased from 31.5 [28.0–37.8] in the pre-pandemic period to 42.5 [35.5–49.5] during the pandemic, a difference that was statistically significant (*p* < 0.001). Of the deaths during the pandemic period, 449 (30.1%) occurred among patients diagnosed with COVID-19 (Table 1).

### 3.2. Antimicrobial Use

#### 3.2.1. Defined Daily Dose (DDD)

During the COVID-19 pandemic, the consumption of several antimicrobial classes increased significantly (Table 2). First- and second-generation cephalosporins showed a non-significant reduction (*p* = 0.061) in DDD/1000 patient-days, with a median of 57.9 (55.37–59.92) to 54.9 (48.30–59.36). Third- to fifth-generation cephalosporins showed a significant increase (*p* = 0.028), from 39.7 (37.45–41.50) to 42.1 (38.16–48.60). Cephalosporins combined with β-lactamase inhibitors increased significantly (*p* < 0.001), rising from 1.2 (0–4.21) to 13.3 (11.12–15.89). Carbapenem use also increased (*p* < 0.001), with medians rising from 27.0 (23.6–29.7) to 30.9 (28.4–34.0), as did glycopeptides (*p* < 0.001), from 39.3 (35.2–43.0) to 49.2 (45.5–56.0).

Time series analysis of DDD, including decomposition into noise, trends, and seasonality (Appendix A), revealed that carbapenem and glycopeptide use varied over time. Both series were non-stationary, showing a continuous upward trend in recent years and recurring seasonal patterns. Significant breakpoints coincided with the COVID-19 pandemic, suggesting its strong influence on the consumption of these antimicrobial classes. Polymyxin B use increased significantly (*p* < 0.001), from 4.6 (3.6–6.39) to 14.4 (8.59–24.69), as did echinocandins (*p* < 0.001), from 17.7 (14.3–20.7) to 26.5 (22.5–31.3).

Analysis of polymyxin B (Appendix A) indicated a significant upward trend over time, with breakpoints in October 2020 and October 2021 suggesting a direct impact of the pandemic on increased consumption. Macrolides (*p* = 0.128) and triazole (*p* = 0.166) use remained stable, while antiviral use decreased significantly (*p* = 0.020), with median falling from 4.1 (3.3–5.9) to 3.1 (1.6–6.2).

#### 3.2.2. Days of Therapy (DOT)

Similarly to DDD, DOT data showed increased antimicrobial use during the pandemic (Appendix A). First- and second-generation cephalosporins decreased significantly (*p* < 0.001), from 75.3 (73.2–80.0) to 68.5 (58.3–73.8) DOT/1000 patient-days, whereas third- to fifth-generation cephalosporins increased (*p* < 0.001), from 38.8 (37.4–40.7) to 43.4 (39.2–50.6). Cephalosporins with β-lactamase inhibitors also increased (*p* < 0.001), from 0.7 (0–3.2) to 11.8 (9.1–14.1). Carbapenem use rose from 31.6 (29.2–34.7) to 38.3 (35.0–40.7) (*p* < 0.001), and glycopeptides from 39.1 (33.8–42.9) to 50.7 (47.8–55.3) (*p* < 0.001). Polymyxin B (*p* < 0.001) increased from 4.3 (3.6–5.4) to 11.1 (6.7–17.9), and echinocandins from 9.6 ± 2.4 to 14.2 ± 4.2 (*p* < 0.001). Triazoles use remained stable (*p* = 0.164), while antivirals decreased significantly (*p* < 0.001), from 8.1 (6.4–9.8) to 4.9 (3.2–7.5).

### 3.3. Healthcare-Associated Infections

Healthcare-associated infections increased significantly during the pandemic period. The incidence rate rose from 1.12 to 2.30 per 10,000 patient-days (*p* < 0.001, Table 3). In raw counts, this corresponded to an increase from 40 cases (22.5%) before the pandemic to 138 cases (77.5%) during the pandemic.

Regarding infections associated with invasive devices, the incidence density of ventilator-associated pneumonia expanded from a mean of 0.7 ± 2.16 before the pandemic to 1.0 ± 1.30 during the pandemic (*p* = 0.016). Central line-associated bloodstream infections rose from 0.2 ± 0.2 to 0.4 ± 0.3, and catheter-associated urinary tract infections from 0.3 ± 0.6 to 0.4 ± 0.4; these differences were not statistically significant (*p* = 0.216 and *p* = 0.453, respectively). ESKAPE group infections also increased during the pandemic. The incidence rate rose from 0.59 to 0.97 per 10,000 patient-days (*p* = 0.076, Table 3). In raw counts, this corresponded to an increase from 21 cases (26.6%) before the pandemic to 58 cases (73.4%).

*Clostridioides difficile* infections declined, with monthly averages decreasing from 9.3 ± 3.5 to 7.3 ± 3.4, median from 9 (6–12.5) to 6 (5–9.5) (*p* = 0.022), and incidence density per 10,000 patient-days from 6.4 ± 2.4 to 4.3 ± 1.8 (*p* < 0.001).

#### Microbiological Profile of Healthcare-Associated Infections

Figure 1 and Figure 2 reveal changes in bacterial resistance profiles in the post-COVID-19 period. There was an increase in cases of multidrug-resistant (MDR) infections, particularly urinary tract infections (UTIs), bloodstream infections (BSIs), and ventilator-associated pneumonia in the ICU (VAP-ICU).

In the pre-pandemic period, MDR infections were relatively limited: *Klebsiella pneumoniae* accounted for five cases and *Pseudomonas aeruginosa* for three cases. Post-pandemic, *K. pneumoniae* increased to 14 MDR cases, representing 29.8% of all the MDR infections, with 40% in BSI and 25% in UTI. *P. aeruginosa* rose to eight MDR cases (17.0%), while *Acinetobacter baumannii*, absent before, appeared in six cases (12.8% of MDR isolates), representing 50% of the MDR VAP-ICU cases.

Other bacteria, including *Enterobacter* spp., showed smaller but notable increases in resistance. MDR profiles also emerged among non-ESKAPE microorganisms, accounting for 34% of total MDR isolates post-pandemic.

### 3.4. Financial Impact

There was an increase in costs during the pandemic period. Median monthly hospital costs rose 39%, from USD 32.4 million (31.2–34.5) to USD 53.2 million (43.8–57.0) (*p* < 0.001). Antimicrobial spending increased by 45.7%, from USD 4.0 million (3.8–4.3) to USD 7.4 million (6.1–8.2) (*p* < 0.001). The median cost per patient increased from USD 7194 (6898–7502) to USD 10,840 (10,317–11,365) (*p* < 0.001), with COVID-19 patients reaching USD 49,660 (33,336–76,132), a fivefold increase (Table 4).

### 3.5. Pharmacist Activities

The median number of monthly pharmaceutical activities increased from 9488 (8703.25–10,102.5) in the pre-pandemic period to 11,589.5 (8061.75–12,504.25) during COVID-19 (*p* = 0.042). Although the median number of activities per pharmacist increased from 214.66 (197.86–230.95) to 250.92 (164.0–262.2), this difference was not statistically significant (*p* = 0.131), suggesting that the individual workload may have varied without a consistent pattern. On the other hand, there was a significant increase in the number of pharmacists per day (*p* < 0.001), with the median rising from 44 (43–44) to 47.5 (45–49).

The number of hospitalized patients per month remained stable (*p* = 0.586) with a similar median: 4524.5 (4265.5–4706.25) before and 4582.5 (3897.25–5272.5) during the pandemic. This indicates that the increased pharmacist workload may be related more to the complexity of care in the context of COVID-19 than the volume of hospitalizations (Table 5).

### 3.6. Adverse Drug Reactions

During the COVID-19 pandemic, there was a significant increase in the total number of adverse drug reactions (ADRs), with 772 cases reported, compared to 312 pre-pandemic cases. The monthly average rose from 13.0 ± 8.0 to 21.4 ± 8.7 (*p* < 0.001). Antimicrobial-related reactions specifically increased from 49 (37%) cases before the pandemic to 81 (63%) during the pandemic.

According to the classification by organ system (Table 6), skin tissue disorders were the most frequent in both periods, representing 30.6% of the ADRs before COVID-19 and 28.4% during COVID-19. Disorders of the renal and urinary systems increased in absolute number from 11 to 15 cases, although the proportion decreased from 22.5% to 18.5%.

A marked reduction in ADRs affecting the lymphatic and blood systems was observed, decreasing from eight cases (16.3%) pre-pandemic to four (4.9%) during the pandemic. Conversely, ADRs of vascular nature increased from 8.2% to 12.4%, and the hepatobiliary ADRs from 2.0% to 7.4% (Table 6).

Before the pandemic, β-lactam antibiotics accounted for most ADRs, particularly in skin (10 cases), hematologic (7 cases), and vascular (3 cases) systems. During the pandemic, although β-lactam antibiotics continued to cause several reactions (16 skin and 5 vascular, among others), ADRs associated with remdesivir emerged, as these drugs were not used pre-pandemic. Remdesivir was associated with renal and urinary (five cases), hepatobiliary (six cases), and immune system (one case) reactions, suggesting a toxicity profile for organs involved in excretion and metabolism. Acyclovir caused nephrotoxicity in 10 cases. In contrast, quinolone-associated ADRs decreased from four to two cases. ADRs involving polymyxins and glycopeptides, previously absent, appeared in small numbers, affecting respiratory, musculoskeletal, and psychiatric systems (Figure 3).

Appendix A shows that ceftriaxone, remdesivir, acyclovir, and cefepime were the drugs most frequently associated with ADRs. Ceftriaxone caused recurrent ADRs both before and during the pandemic, mainly affecting the skin, hematologic, and vascular systems. Cefepime predominantly caused skin and hematologic reactions in the pre-pandemic period. Other drugs, such as cefazolin, clindamycin, and polymyxin B, were less frequently associated with ADRs, affecting the skin, nervous system, and musculoskeletal system.

### 3.7. Linear Regression Models

Linear regression models evaluated the factors associated with the mean defined daily dose (DDD) by antimicrobial class. Three covariates showed statistically significant associations across classes: total patient-days, total hospital costs, and COVID-19 deaths. Positive coefficients indicate higher DDD (greater consumption), whereas negative coefficients indicate lower consumption.

Carbapenem (Appendix A) and glycopeptide (Appendix A) use showed a positive association with total hospital cost. For macrolides, mean DDD was positively associated with COVID-19 deaths and adverse drug reactions (ADRs) reports (Appendix A). Polymyxin consumption was also positively associated with the number of COVID-19 deaths (Appendix A). Echinocandin DDD increased due to higher hospital costs, multidrug resistance incidence, and pharmaceutical interventions (Appendix A). Antiviral use was positively associated with ADR reports (Appendix A).

## 4. Discussion

This study highlighted the multiple impacts of the COVID-19 pandemic on hospital dynamics, with changes in clinical care, microbiological profiles, therapeutic practices, and financial indicators. There was a significant increase in hospitalizations, deaths, healthcare-associated infections (HAIs), antimicrobial consumption, and hospital costs. Furthermore, high rates of adverse drug reactions (ADRs) and a significant burden on healthcare professionals, particularly clinical pharmacists, were recorded. The complexity of care imposed by the pandemic reinforced the demand to adopt robust epidemiological surveillance strategies and rational antimicrobial management, especially in health crisis contexts.

Although the rise in hospitalizations during the pandemic did not reach statistical significance, the observed trend suggests relevant care pressures, which may have contributed to changes in the dynamics of hospital admissions and patient flows [18]. Hospital mortality increased significantly, and of the deaths recorded during the pandemic, 449 (30.1%) occurred in patients diagnosed with COVID-19. This increase is consistent with data in the literature, which indicates higher hospital mortality rates during pandemic peaks, often associated with bed shortages, limited response capacity, and overloaded health services [19]. In a study by Piroth et al., the mortality rate for COVID-19 (16.9%) was significantly higher than that observed for seasonal influenza (5.8%) during the 2018–2019 season, a difference attributed to the greater clinical aggressiveness of COVID-19, including inflammatory and thromboembolic complications, and the initial lack of effective therapies [20].

Across antimicrobial classes, DDD and DOT for carbapenems, glycopeptides, polymyxin B, and echinocandins rose, with surges overlapping with pandemic peaks. This behavior, also documented in other studies [21,22], was influenced by the empirical management of severe COVID-19 cases and the initial difficulty in differentiating pure viral infections from bacterial coinfections. Data support this scenario: there was an increase in the use of broad-spectrum β-lactams (from 78.1 to 142.5 DDD/1000 patient-days), carbapenems (from 50.9 to 110.1), and colistin (from 4.1 to 13.3), all with *p* < 0.001 [23,24,25].

Carbapenems and glycopeptides were closely associated with greater clinical severity and correlated with increased hospital costs. This pattern reflects their use in empiric therapy for suspected bacterial coinfections in critically ill patients, often without microbiological confirmation [26]. While understandable in the context of clinical uncertainty, such practices underscore the importance of antimicrobial stewardship programs.

The use of polymyxin B, a last-line antimicrobial, also increased significantly, with mean DDD positively associated with COVID-19 deaths. These findings demonstrate its role in treating infections caused by multidrug-resistant pathogens. Similar patterns have been described in Europe and Asia, where increased polymyxin consumption was reported in ICUs with high rates of resistant Gram-negative organisms, such as *Acinetobacter baumannii* and *Pseudomonas aeruginosa* [27].

Echinocandin use correlated with higher costs, a greater number of pharmaceutical interventions, invasive fungal infections, and multidrug resistance. This demonstrates that echinocandins were primarily used in complex cases, such as severe candidemia or coinfections in immunosuppressed patients [28,29].

Macrolides, particularly azithromycin, were widely prescribed early in the pandemic due to their presumed anti-inflammatory and antiviral properties. However, subsequent studies questioned their effectiveness in the absence of bacterial infection, and their use became associated with increased COVID-19 mortality and adverse drug responses [26,27].

The use of antivirals showed a significant decline during the pandemic. The antivirals included in this study were ganciclovir, remdesivir, acyclovir, and oseltamivir. The reduction can be explained by protocol revisions and clinical practice adaptations. For example, oseltamivir was abandoned after being deemed ineffective for COVID-19, while ganciclovir and acyclovir were used less frequently due to their narrower clinical indications. Remdesivir emerged as a new therapeutic option but was also linked to renal, hepatic, and immune-related ADRs. Notably, Paxlovid was not included in this study, as it became available only later. This downward trend in antiviral use was also observed in other countries, including the United Kingdom and Australia, where the abandonment of ineffective therapies and revisions of clinical protocols resulted in reduced utilization [30]. The negative correlation between antiviral use and hospital costs may reflect budgetary constraints or substitution with more effective therapies.

Triazole antifungal use remained stable, while overall antiviral use decreased sharply, reinforcing the broader trend of abandoning ineffective treatments and adapting care protocols in light of emerging evidence.

These changes in prescribing patterns reflect the overload faced by hospital systems and, importantly, the partial suspension or bypassing of structured antimicrobial stewardship programs during the health crisis. The shift in antimicrobial classes was driven by a combination of factors: the clinical challenge of differentiating severe COVID-19 from bacterial coinfections due to overlapping symptoms (e.g., fever and pulmonary infiltrates), which led to empirical broad-spectrum use; a perceived need to avoid initial treatment failure in critically ill patients, often leading to the choice of last-resort agents like carbapenems and polymyxins; and specific, albeit decentralized, changes in clinical practice as traditional guidelines were supplanted by emergent, often informal, consensus among overwhelmed frontline teams. This period demonstrated that the COVID-19 pandemic directly challenged the core principles of antimicrobial stewardship, leading to increased antimicrobial use in the absence of well-structured guidelines [31,32].

At the same time, HAIs increased, particularly those associated with invasive devices such as mechanical ventilators, central venous catheters, and urinary catheters. This aligns with international studies linking higher HAI rates to hospital overload and prioritization of emergency measures over routine infection control [33].

Microbiologically, the study observed rising rates of multidrug-resistant organisms, including increased *Klebsiella pneumoniae*, the emergence of multidrug-resistant *Acinetobacter baumannii*, and the higher prevalence of *Pseudomonas aeruginosa*. This resurgence of antimicrobial resistance, occurring concurrently with the documented relaxation of antimicrobial stewardship, demonstrates a direct consequence of the extensive empirical and broad-spectrum prescribing that characterized the pandemic response. These trends suggest a reversal of prior gains in antimicrobial resistance control and align with reviews reporting 15–78% increases in resistant pathogens among COVID-19 patients, with a subsequent impact on mortality [2].

Conversely, a statistically significant reduction in *Clostridioides difficile* infections was recorded. This may be explained by more stringent infection control measures, such as patient isolation, personal protective equipment, and enhanced sanitation, adopted in hospitals dedicated exclusively to COVID-19 care, which appear to have offset the increased antimicrobial use [34,35].

The pandemic also broadened the scope of work for hospital pharmacists, necessitating team reorganization, expanded responsibilities, and adaptations in internal processes to ensure continuity of care. The findings of this study emphasize the need for workforce expansion to adequately meet the demands arising from increased activity and care complexity [36]. Paudyal et al., in a study across 16 European countries, confirmed this trend, highlighting accelerated adoption of remote practices and the importance of continuous training [37]. Pharmacists increasingly contributed to infection control, antimicrobial management, clinical decision-making, and therapeutic protocol review, while also addressing drug shortages and heightened demands for pharmacovigilance [38,39].

The intensification of care, greater use of high-cost medications, and the need for advanced supportive therapies significantly increased hospitalization costs. International data corroborate these findings: in the United States, the mean cost per inpatient stay rose from USD 10,394 in March 2020 to USD 13,072 by March 2022, with even higher costs for advanced therapies such as ECMO [40], while in Europe, COVID-19-related costs were especially high in intensive care units [41].

The observed increase in ADRs is consistent with reports linking intensive pharmacological exposure and systemic inflammation in COVID-19 to greater adverse event risks. Skin reactions remained the most common, especially with β-lactams, while hepatobiliary and vascular ADRs increased. Remdesivir was strongly associated with renal, hepatic, and immunological reactions, highlighting the need for pharmacovigilance. The diversity of ADRs observed underscores the risks of extensive empirical antimicrobial use and the importance of active monitoring and cautious prescribing [42,43].

This study has several limitations. Its single-center design limits the generalizability of the findings to other settings. Furthermore, the retrospective observational design prevents the establishment of causal relationships. While the pandemic is a major explanatory factor for the observed changes, our study design cannot exclude the influence of other concurrent factors. These may include shifts in infection prevention and control measures, and adaptations in local treatment policies, all of which could have independently affected antimicrobial use and other outcomes. Another limitation is the use of all the hospitalized patients as the denominator for antimicrobial consumption. The correct denominator for assessing the appropriateness of antibiotic use would be patients with infections. Since we cannot ascertain whether the proportion of patients with bacterial infections changed during the pandemic, our ability to draw definitive conclusions about the drivers of antimicrobial consumption is limited. We could only track device-associated infections, which do not represent the total burden of infections in the hospital. Furthermore, our analysis was unable to assess the rate of multiple concurrent antibiotic prescriptions per patient, a valuable metric for understanding prescribing complexity, as our data focused on aggregate consumption. The absence of post-pandemic (post-2022) data means that we cannot yet assess whether the identified trends have persisted or normalized. While we used standardized metrics, the retrospective nature of the data collection is subject to information bias. We attempted to mitigate analytical bias by using appropriate time-series analysis and checking regression model assumptions. The cost analysis focused on overall hospitalization and antimicrobial use without stratification by department, materials, or procedures. Despite these limitations, the study offers several strengths. It integrates multiple outcomes related to COVID-19 to provide a comprehensive view of the pandemic’s hospital impacts. Its differentiated statistical approach identified significant correlations between variables over time, strengthening the validity of the findings. The results provide useful information for institutional policies on antimicrobial stewardship programs, epidemiological surveillance, and sustainable financial planning in health crises.

## 5. Conclusions

This study documents a significant association between the COVID-19 pandemic period and profound changes in a hospital ecosystem, including increased antimicrobial consumption, resistance rates, and costs. The challenges observed should not be regarded solely as inevitable consequences of a health crisis, but as an important lesson on the vulnerability of healthcare systems. Our findings underscore the urgent need to strengthen integrated surveillance; reinforce antimicrobial stewardship programs, especially during emergencies; and build more resilient, multidisciplinary teams to mitigate the impact of future crises, regardless of their specific cause.

## Figures and Tables

**Figure 1 pathogens-14-01126-f001:**
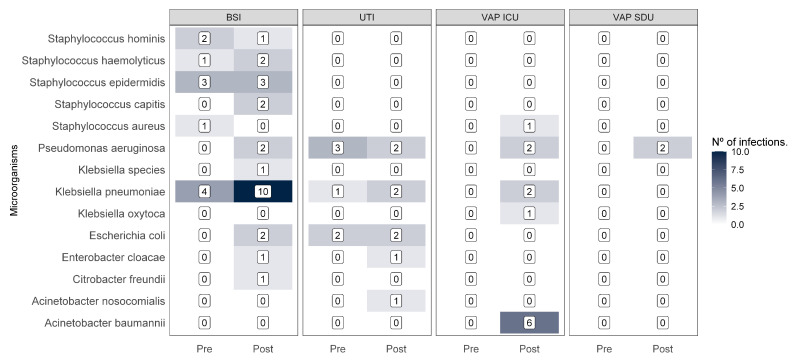
Heatmaps of multidrug-resistant microorganisms at different infection sites before and during the pandemic. Note: BSI: bloodstream infection; UTI: urinary tract infection; VAP-SDU: ventilator-associated pneumonia in the step-down unit; VAP-ICU: ventilator-associated pneumonia in the intensive care unit.

**Figure 2 pathogens-14-01126-f002:**
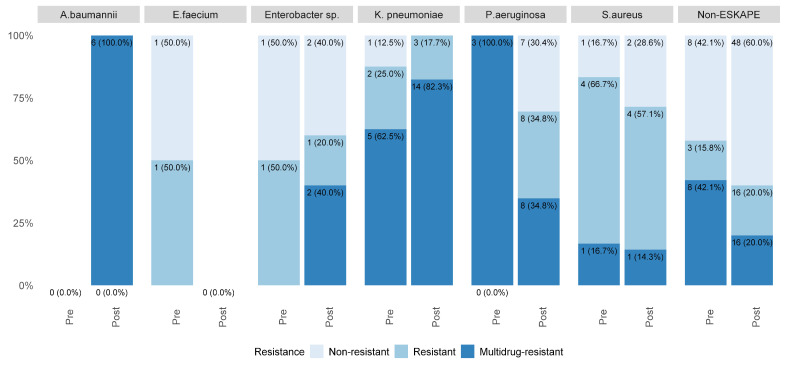
Microbiological profile regarding antimicrobial resistance before and during the COVID-19 pandemic.

**Figure 3 pathogens-14-01126-f003:**
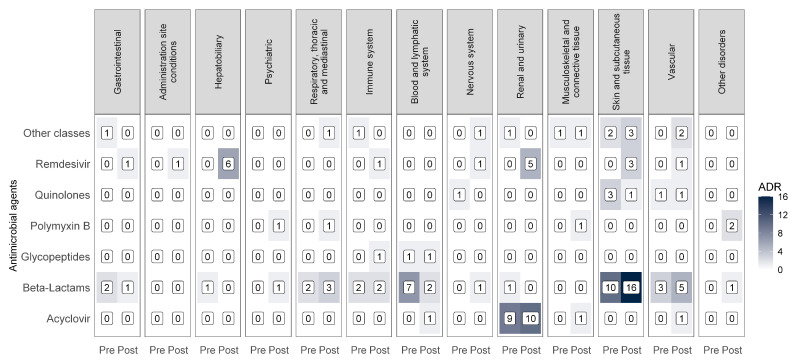
Heatmap of adverse reactions by antimicrobial class before and during COVID-19.

**Table 1 pathogens-14-01126-t001:** A comparison of hospitalization data, antimicrobial use, and deaths before and during the COVID-19 pandemic.

	Before COVID-19	During COVID-19	*p*-Value
	N (%)	N (%)	
**Hospitalized patients**			
Total	116,591 (43.4%)	152,122 (56.6%)	
Monthly mean (±SD)	4484 (±286)	4474 (±874)	
Monthly Median [IQR]	4525 [4266–4706]	4.583 [3897–5273]	0.586
**Hospitalized patients using antimicrobials**			
Total	60,502 (42.8%)	80,922 (57.2%)	
Monthly mean (±SD)	2327 (±149.1)	2380 (±464.4)	
Monthly Median [IQR]	2334 [2257–2459]	2411 [2121–2620]	0.399
**Patient/days**			
Total	415,252 (43.3%)	543,500 (56.7%)	
Monthly mean (±SD)	15,971 (±1915.9)	15,985 (±2325.2)	
Monthly Median [IQR]	15,971 [14,426–18,087]	15,860 [14,619–17,516]	0.706
**Hospitalized patients with COVID-19 using antimicrobials**	-	6936	-
**Deaths**			
Total	861 (36.6%)	1494 (63.4%)	
Monthly mean (±SD)	33.12 (±7.7)	43.94 (±13.0)	
Monthly Median [IQR]	31.5 [28.0–37.8]	42.5 [35.5–49.5]	<0.001
**Deaths in COVID-19 patients**	-	449 (30.1%)	-

Note: n = number; SD: standard deviation; IQR: interquartile range (1st and 3rd quartiles); *p*-value = Mann–Whitney test.

**Table 2 pathogens-14-01126-t002:** Antimicrobial consumption (DDD) per 1000 patient-days comparison before and during the COVID-19 pandemic.

Antimicrobial	Before COVID-19	During COVID-19	*p*-Value
**1st and 2nd generation cephalosporin (cefazolin, cephalothin, cefuroxime)**			
Mean ± SD	57.2 ± 5.2	52.1 ± 10.4	
Median [IQR]	57.8 [55.3–59.9]	54.8 [48.3–59.3]	0.061
**3rd, 4th, and 5th cephalosporin (ceftriaxone, cefotaxime, ceftazidime, cefepime, ceftaroline)**			
Mean ± SD	39.7 ± 2.7	44.2 ± 7.6	
Median [IQR]	39.7 [37.4–41.5]	42.1 [38.1–48.60]	0.028
**Cephalosporin + β-lactamase inhibitors**			
Mean ± SD	2.4 ± 2.8	13.2 ± 4.6	
Median [IQR]	1.1 [0–4.2]	13.3 [11.1–15.8]	<0.001
**Macrolides**			
Mean ± SD	20.1 ± 3.3	28.1 ± 15.8	
Median [IQR]	19.3 [18.0–22.7]	24.3 [16.5–32.5]	0.128
**Carbapenems**			
Mean ± SD	26.8 ± 4.0	31.4 ± 4.4	
Median [IQR]	26.9 [23.6–29.7]	30.8 [28.4–34.0]	<0.001
**Glycopeptides**			
Mean ± SD	39.2 ± 4.8	52.5 ± 11.4	
Median [IQR]	39.2 [35.2–43.0]	49.1 [45.5–56.0]	<0.001
**Polymyxin B**			
Mean ± SD	5.1 ± 2.9	19.1 ± 14.7	
Median [IQR]	4.6 [3.6–6.3]	14.4 [8.5–24.6]	<0.001
**Echinocandins**			
Mean ± SD	17.0 ± 5.1	28.4 ± 9.6	
Median [IQR]	17.6 [14.3–20.7]	26.5 [22.5–31.3]	<0.001
**Triazole Antifungals**			
Mean ± SD	15.5± 3.3	17.0 ± 4.6	0.166 *
Median [IQR]	15.7 [13.2–18.0]	16.2 [14.0–19.8]	
**Antivirals**			
Mean ± SD	8.8 ± 10.9	4.8 ± 6.1	
Median [IQR]	4.1 [3.3–5.9]	3.0 [1.6–6.2]	0.020

Note: IQR: interquartile range (1st and 3rd quartiles); SD: standard deviation. *p*-value = Mann–Whitney test or * Student’s *t*-test.

**Table 3 pathogens-14-01126-t003:** Healthcare-associated infections before and during the COVID-19 pandemic.

	Total Before COVID-19	Before COVID-19	Total During COVID-19	During COVID-19	*p*-Value
	N (%)		N (%)		
**Healthcare-associated infections**	40 (22.5%)		138 (77.5%)		
**Healthcare-associated infections incidence rate per 10,000 patient-days**	-	1.12	-	2.30	<0.001
**Central line-associated bloodstream infections**	27 (24.5%)		83 (75.5%)		
**Bloodstream infection incidence rate per 1000 catheter-days**					
Mean ± SD		0.28 ± 0.29		0.40 ± 0.32	
Median [IQR]		0.25 [0.00–0.49]		0.34 [0.16–0.61]	0.216
**Catheter-associated urinary tract infections**	9 (31.0%)		20 (69.0%)		
**Urinary tract infection incidence rate per 1000 catheter-days**					
Mean ± SD		0.35 ± 0.61		0.40 ± 0.49	
Median [IQR]		0.00 [0.00–0.83]		0.00 [0.00–0.73]	0.453
**Ventilator-associated pneumonia**	4 (10.3%)		35 (89.7)		
**Ventilator-associated pneumonia incidence rate per 1000 ventilator-days**					
Mean ± SD		0.72 ± 2.16		1.08 ± 1.30	
Median [IQR]		0.0 [0.00–0.00]		0.35 [0.00–1.97]	0.016
**Group ESKAPE infections**	21 (26.6%)		58 (73.4%)		
**Group ESKAPE infections incidence rate per 10,000 patient-days**	-	0.59	-	0.97	0.076

Note: IQR: interquartile range (1st and 3rd quartiles); SD: standard deviation. *p*-value = Mann–Whitney test.

**Table 4 pathogens-14-01126-t004:** Hospital financial impact before and during COVID-19 pandemic.

	Before COVID-19	During COVID-19	Change (%)	*p*-Value
**Total Costs (millions of US$)**	844.52	1682.93		-
Median monthly [IQR]	32.42 [31.22–34.47]	53.20 [43.82–56.96]	39.0%	<0.001
Range	25.82–38.16	20.49–61.71		
**Monthly antimicrobial costs (millions of US$)**				
Median [IQR]	4.03 [3.76–4.34]	7.44 [6.11–8.22]	45.7%	<0.001
Range	3.04–5.60	3.37–9.46		
**Cost per patient (US$)**				
Median [IQR]	7194 [6898–7502]	10,840 [10,317–11,365]	33.6%	<0.001
Range	6037–8635	8311–16,750		
**Total costs of COVID-19 patients (millions of US$)**				
Median [IQR]	-	8.66 [5.56–12.03]	-	
Range	-	0.21–22.77	-	
**Cost per patient with COVID-19 (US$)**				
Median [IQR]	-	49,660 [33,336–76,132]	-	
Range	-	5302–135,866	-	

Note: IQR: interquartile range (1st and 3rd quartiles). *p*-value = Mann–Whitney test.

**Table 5 pathogens-14-01126-t005:** Workload for pharmacist’s comparison before and during COVID-19.

	Before COVID-19	During COVID-19	*p*-Value
**Monthly pharmaceutical activities**			
Median [IQR]	9488 [8703.25–10,102.5]	11,589.5 [8061.75–12,504.25]	0.042
Range	7072–11,324	6596–14,519	
**Monthly pharmaceutical activities per pharmacist**			
Median [IQR]	214.66 [197.86–230.95]	250.92 [164.07–262.25]	0.131
Range	164.5–248.6	140.3–312.4	
**N of pharmaceutics/day**			
Median [IQR]	44 [43–44]	47.5 [45–49]	<0.001
Range	42–47	43–51	
**Hospitalized patients/month**			
Median [IQR]	4524.5 [4265.5–4706.25]	4582.5 [3897.25–5272.5]	0.586
Range	4002–4948	1979–5565	

Note: IQR: interquartile range (1st and 3rd quartiles); *p*-value = Mann–Whitney test.

**Table 6 pathogens-14-01126-t006:** Adverse drug reactions (ADRs) to antimicrobials according to the system organ.

System	Total ADRs	Before COVID-19	During COVID-19
Skin tissue	38	15 (30.61%)	23 (28.40%)
Renal and urinary	26	11 (22.45%)	15 (18.52%)
Lymphatic and blood *	12	8 (16.33%)	4 (4.94%)
Vascular *	14	4 (8.16%)	10 (12.35%)
Immune	7	3 (6.12%)	4 (4.94%)
Hepatobiliary	7	1 (2.04%)	6 (7.41%)
Gastrointestinal	5	3 (6.12%)	2 (2.47%)
Respiratory	7	2 (4.08%)	5 (6.17%)
Nervous	4	1 (2.04%)	3 (3.70%)
Connective tissue and musculoskeletal	4	1 (2.04%)	3 (3.70%)
Psychiatric	2	0 (0%)	2 (2.47%)
Others	4	0 (0%)	4 (4.94%)

Note: * Examples of ADRs: Lymphatic and blood system—agranulocytosis (e.g., due to beta-lactams); Vascular system—vasculitis (e.g., drug-induced hypersensitivity vasculitis).

## Data Availability

The original contributions presented in the study are included in the article/Appendix A. Further inquires can be directed to the corresponding author.

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
