# Peer review of "Beyond the Virus: The Collateral Impact of COVID-19 on Antimicrobial Consumption, Microbial Resistance, and Pharmacoeconomics"

_pathogens, 2025, doi:10.3390/pathogens14111126_

Round 1

Reviewer 1 Report

Comments and Suggestions for Authors

General Comments

This is a well structured manuscript that explores the effects of COVID-19 pandemic on antimicrobial use, resistance patterns, and pharmaco-economic outcomes in a large tertiary Brazilian hospital. The findings are relevant , the English is clear and the discussion is ok, but several revisions are necessary to enhance scientific rigor, reproducibility, and contextual strength. Authors may stated if they used AI (some parts it appears to be written with an AI)

Major Comments:

  1. Introduction

    The introduction would benefit from an integration of recent literature on antibiotic use during the COVID-19 pandemic. It currently describes the issue in general terms without providing data-based context.

    I recommend including data from recent meta-analyses (such as 10.3390/jcm11237207) and incorporating it in "Introduction" section.

    Additionally, it would be useful to clarify in Introduction whether the study addresses all antimicrobial classes (i.e. antibacterials, antifungals, and antivirals) or primarily on antibacterial drugs. 

  1. Materials and Methods

    This section is detailed, but it includes some narrative sentences that should be removed or relocated to supplementary materials

    For example, the following passage should be omitted from the main text and relocated to the supplementary appendix:
    “The hospital employs approximately 50 clinical pharmacists across multiple specialties. The term ‘number of pharmaceutical activities’ refers to the total count of all documented professional actions performed by clinical pharmacists. This measure serves as a proxy for total clinical workload volume, representing the sum of all recorded interventions without specifying their individual nature, thus reflecting the overall intensity of pharmacy services.”

    Moreover, the authors should state which tests were used for this categorical variable -if one- (e.g., Fisher’s exact test or Chi-square test) and under what conditions each was applied.

    Finally, authors should state how the normality test of quantitative variables was assessed prior to choosing between parametric (t-test) and non-parametric (Mann–Whitney U) comparisons.

  2. Language and Style

The manuscript is well written, but it appears that AI may have provided some help. Furthermore, in the Materials and Methods and Discussion authors may avoid redundancy (i.e. “This reflects…” or “This suggests…” )

Author Response

REVIEWER 1

Open Review

Quality of English Language

(x) The English could be improved to more clearly express the research.
( ) The English is fine and does not require any improvement.

Yes

Can be improved

Must be improved

Not applicable

Does the introduction provide sufficient background and include all relevant references?

( )

( )

(x)

( )

Is the research design appropriate?

( )

(x)

( )

( )

Are the methods adequately described?

( )

(x)

( )

( )

Are the results clearly presented?

( )

(x)

( )

( )

Are the conclusions supported by the results?

( )

(x)

( )

( )

Are all figures and tables clear and well-presented?

( )

(x)

( )

( )

Comments and Suggestions for Authors

General Comments

This is a well structured manuscript that explores the effects of COVID-19 pandemic on antimicrobial use, resistance patterns, and pharmaco-economic outcomes in a large tertiary Brazilian hospital. The findings are relevant , the English is clear and the discussion is ok, but several revisions are necessary to enhance scientific rigor, reproducibility, and contextual strength. Authors may stated if they used AI (some parts it appears to be written with an AI)

Response: We appreciate your comments.

Major Comments:

  1. Introduction

The introduction would benefit from an integration of recent literature on antibiotic use during the COVID-19 pandemic. It currently describes the issue in general terms without providing data-based context.

I recommend including data from recent meta-analyses (such as 10.3390/jcm11237207) and incorporating it in "Introduction" section.

Additionally, it would be useful to clarify in Introduction whether the study addresses all antimicrobial classes (i.e. antibacterials, antifungals, and antivirals) or primarily on antibacterial drugs. 

Response: Thank you very much for the comments. We have incorporated recent data-based evidence from the systematic review (10.3390/jcm11237207 – Granata et al. – Reference#7) to strengthen the Introduction with current findings on antibiotic use during the COVID-19 pandemic. Furthermore, we have clarified in the Introduction that the study primarily focuses on antibacterial drug use, while antifungal and antiviral utilization are also considered where relevant to provide a comprehensive overview of antimicrobial stewardship practices. These revisions have been added in the Introduction section, paragraphs 5 and 7, respectively.

  1. Materials and Methods

This section is detailed, but it includes some narrative sentences that should be removed or relocated to supplementary materials

For example, the following passage should be omitted from the main text and relocated to the supplementary appendix:
“The hospital employs approximately 50 clinical pharmacists across multiple specialties. The term ‘number of pharmaceutical activities’ refers to the total count of all documented professional actions performed by clinical pharmacists. This measure serves as a proxy for total clinical workload volume, representing the sum of all recorded interventions without specifying their individual nature, thus reflecting the overall intensity of pharmacy services.”

Response: We appreciate the reviewer's comment to improve conciseness. We have removed the sentence "The hospital employs approximately 50 clinical pharmacists across multiple specialties" from the main text. However, we have retained the definition of "number of pharmaceutical activities" because it provides essential context for interpreting this key metric in the results and discussion sections. We hope this balanced revision addresses the reviewer’s concern.

Moreover, the authors should state which tests were used for this categorical variable -if one- (e.g., Fisher’s exact test or Chi-square test) and under what conditions each was applied.

Response: Thank you for this feedback. We have now explicitly stated the statistical tests and the conditions for applying each in the “Statistical analysis” (one pages 3 and 4). Since we did not use the Chi-square and Fisher's tests in the manuscript's results, they were not included in the statistical analysis.

Finally, authors should state how the normality test of quantitative variables was assessed prior to choosing between parametric (t-test) and non-parametric (Mann–Whitney U) comparisons.

Response: Data normality was assessed using the Shapiro-Wilk test. The normality of quantitative variables was assessed using the Shapiro-Wilk test. Based on this assessment, the comparison of outcome variables between the two periods was performed using the Student's t-test for normally distributed data or the Mann-Whitney U test for non-normally distributed data. We have clarified this in the first paragraph of page 4.

  1. Language and Style

The manuscript is well written, but it appears that AI may have provided some help. Furthermore, in the Materials and Methods and Discussion authors may avoid redundancy (i.e. “This reflects…” or “This suggests…” )

Response: We thank the reviewer for the feedback on the manuscript's style. We can confirm that no AI writing tools were used in the preparation of this manuscript. The observed stylistic patterns, including phrases like "This reflects...", likely stem from the fact that this work is derived from a master's thesis originally written in Portuguese by a team of native Portuguese speakers. For instance, the word “reflects” was used once”, and “suggests” was used three times. We acknowledge the reviewer's point on redundancy and have worked with our co-author, Prof. Michael B. Edmond, to refine the language and minimize these repetitions in the revised manuscript.

Reviewer 2 Report

Comments and Suggestions for Authors

This manuscript is a well-thought-out and thorough examination of antimicrobial use in a hospital system during the pandemic. The breakdown and statistical comparison between multiple factors involved with patient care, treatment and relevant clinical parameters is thorough and thought-provoking. In that sense, I would have appreciated the author's insights into the reasons for shifting antimicrobial in the discussion or conclusion statements. For example, was it clinically based on overlapping symptoms with COVID, perceived avoidance of AMR with certain classes of drugs or other specific changes in hospital directives, guidelines or policies?  These are alluded to in the discussion, but the specific guideline with the resulting change in AMR or clinical outcome is absent. Figures could be improved by increasing the size of fonts, Figure 2 could be improved by providing "pre" and "post" for each joined drug side by side, rather than in two left and right plots, to reduce difficulty in comparing percentages pre and post for each microbial strain. Another recommendation is having an abbreviations list. A question worth consideration is the rate of multiple antibiotic prescriptions per case or patient day, up/down or same? Data for the number of clinical isolates for each strain to be stated would make the changes in AMR easier to appreciate. It is discussed this way for Klebsiella and HAI. 

Author Response

REVIEWER 2

Open Review

Quality of English Language

( ) The English could be improved to more clearly express the research.
(x) The English is fine and does not require any improvement.

Yes

Can be improved

Must be improved

Not applicable

Does the introduction provide sufficient background and include all relevant references?

(x)

( )

( )

( )

Is the research design appropriate?

(x)

( )

( )

( )

Are the methods adequately described?

(x)

( )

( )

( )

Are the results clearly presented?

(x)

( )

( )

( )

Are the conclusions supported by the results?

( )

(x)

( )

( )

Are all figures and tables clear and well-presented?

( )

(x)

( )

( )

Comments and Suggestions for Authors

This manuscript is a well-thought-out and thorough examination of antimicrobial use in a hospital system during the pandemic. The breakdown and statistical comparison between multiple factors involved with patient care, treatment and relevant clinical parameters is thorough and thought-provoking. In that sense, I would have appreciated the author's insights into the reasons for shifting antimicrobial in the discussion or conclusion statements. For example, was it clinically based on overlapping symptoms with COVID, perceived avoidance of AMR with certain classes of drugs or other specific changes in hospital directives, guidelines or policies?  These are alluded to in the discussion, but the specific guideline with the resulting change in AMR or clinical outcome is absent.

Response: We appreciate your comments. In response, we have revised the discussion to provide a more explicit analysis of the specific reasons behind the observed shifts in antimicrobial prescribing. We now clearly state that the changes were driven by a combination of the clinical difficulty in distinguishing severe COVID-19 from bacterial co-infections, a perceived need to avoid initial treatment failure in critically ill patients, and decentralized changes in practice as formal guidelines were bypassed in favor of emergent consensus among frontline teams. This addition can be found on page 13.

We agree that connecting the changes in practice to the antimicrobial resistance outcomes strengthens our analysis. We have added the following sentence to the AMR paragraph on page [13], lines [411-414]: This resurgence of antimicrobial resistance, occurring concurrently with the documented relaxation of antimicrobial stewardship, suggests a direct consequence of the extensive empirical and broad-spectrum prescribing that characterized the pandemic response.”.

Figures could be improved by increasing the size of fonts, Figure 2 could be improved by providing "pre" and "post" for each joined drug side by side, rather than in two left and right plots, to reduce difficulty in comparing percentages pre and post for each microbial strain. Another recommendation is having an abbreviations list.

Response: Thank you for your suggestions. We have changed the figures by increasing the font size. We have replaced Figure 2 as you suggested. We have also added a list of abbreviations at the end of the manuscript.

A question worth consideration is the rate of multiple antibiotic prescriptions per case or patient day, up/down or same?

Response: Regarding the rate of multiple antibiotic prescriptions, we agree this is a valuable metric. Unfortunately, this specific data point (e.g., simultaneous prescriptions per patient-day) was not captured in our dataset, which focused on aggregate consumption (DDD, DOT). We have acknowledged this limitation in the revised manuscript on page [14], lines [450-453]: “Furthermore, our analysis was unable to assess the rate of multiple concurrent antibiotic prescriptions per patient, a valuable metric for understanding prescribing complexity, as our data focused on aggregate consumption.”.

Data for the number of clinical isolates for each strain to be stated would make the changes in AMR easier to appreciate. It is discussed this way for Klebsiella and HAI. 

Response: Thank you for this suggestion. We have revised the text to explicitly report the number of clinical isolates for each bacterial strain in both pre- and post-pandemic periods (on section 3.3.1).

Reviewer 3 Report

Comments and Suggestions for Authors

The study titled ‘Beyond the Virus: The Collateral Impact of COVID-19 on Anti-microbial Consumption, Microbial Resistance, and Pharmacoeconomics.’ investigated the effect of the COVID-19 pandemic on antimicrobial use in hospitalized patients focusing on therapeutic, microbiological and pharmacoeconomic aspects. The study is a single centre retrospective observational study and was carried out in one Brazilian hospital.

Comments:

  • The main limitation of the study is its retrospective single-centre design. This design does not allow drawing conclusions on causal effects. However, the authors describe the pandemic as the cause for any observed difference. One cannot exclude that other factors caused the observed changes, such as changes infection prevention and control measures, treatment policies, patient mix, etc. There might be more patients with (bacterial) infectious diseases (including HAIs) which require antibiotics during the pandemic. Therefore, I suggest that the authors revise their conclusions. There is an association between the pandemic and antimicrobials use but other factors, independent of the pandemic may also had effects.  
  • Moreover, I suggest that the authors report their findings in a more balanced fashion. For example, the authors show that the mean monthly number of patients receiving antimicrobials showed no significant changes. Reporting raw counts is not meaningful due to the longer pandemic period. I suggest showing these numbers in a Table but I would remove these results (raw counts) from the text, especially from the abstract.
  • I suggest that the authors present the numbers of patients who were hospitalized due to infectious diseases and numbers for patients with HAIs. These numbers are more relevant than the absolute numbers of hospitalized patients. All outcomes for antimicrobial use should be reported in relation to the number of hospitalized patients with infectious diseases and HAIs. In the present form, all numbers are presented in relation to all hospitalized patients, which is not appropriate.
  • The study would strongly benefit if the authors would include data from the post-pandemic (i.e. after 2022). It would be interesting to see whether the observed effects during the pandemic may have ‘normalized’ to pre-pandemic numbers. These results would actually be much more interesting than the current limited analysis. There are already dozens of studies reporting the effect of the pandemic on antimicrobial use and infectious disease prevalence.
  • The discussion of the limitations is very weak. This study has multiple limitations that should be discussed, including their effects on the results. Some limitations are the single-centre design, the observational design, the retrospective design, and the lack of ‘before’ and ‘after’ data that allow a comparison of the patient populations.
  • The authors should revise their manuscript according to the STROBE guidelines (STrengthening the Reporting of OBservational studies in Epidemiology). Please provide the checklist with the revised manuscript.
  • The caption of Table 3 is wrong.
  • It is unclear whether the hospital is a tertiary or quaternary referral hospital. The authors use both terms in their manuscript.

Author Response

REVIEWER 3

Open Review

Quality of English Language

( ) The English could be improved to more clearly express the research.
(x) The English is fine and does not require any improvement.

Yes

Can be improved

Must be improved

Not applicable

Does the introduction provide sufficient background and include all relevant references?

(x)

( )

( )

( )

Is the research design appropriate?

( )

( )

(x)

( )

Are the methods adequately described?

( )

(x)

( )

( )

Are the results clearly presented?

(x)

( )

( )

( )

Are the conclusions supported by the results?

( )

( )

(x)

( )

Are all figures and tables clear and well-presented?

(x)

( )

( )

( )

Comments and Suggestions for Authors

The study titled ‘Beyond the Virus: The Collateral Impact of COVID-19 on Anti-microbial Consumption, Microbial Resistance, and Pharmacoeconomics.’ investigated the effect of the COVID-19 pandemic on antimicrobial use in hospitalized patients focusing on therapeutic, microbiological and pharmacoeconomic aspects. The study is a single centre retrospective observational study and was carried out in one Brazilian hospital.

Response: We appreciate your comments.

Comments:

The main limitation of the study is its retrospective single-centre design. This design does not allow drawing conclusions on causal effects. However, the authors describe the pandemic as the cause for any observed difference. One cannot exclude that other factors caused the observed changes, such as changes infection prevention and control measures, treatment policies, patient mix, etc. There might be more patients with (bacterial) infectious diseases (including HAIs) which require antibiotics during the pandemic. Therefore, I suggest that the authors revise their conclusions. There is an association between the pandemic and antimicrobials use but other factors, independent of the pandemic may also had effects.  

Response: We thank the reviewer for this comment. We agree that our single-center, retrospective design cannot establish causality. We have revised the Limitations and Conclusions sections to explicitly state that the pandemic is an associative, rather than a definitive causal, factor and that other concurrent variables may have contributed to the observed changes.

Moreover, I suggest that the authors report their findings in a more balanced fashion. For example, the authors show that the mean monthly number of patients receiving antimicrobials showed no significant changes. Reporting raw counts is not meaningful due to the longer pandemic period. I suggest showing these numbers in a Table but I would remove these results (raw counts) from the text, especially from the abstract.

Response: Thank you for this suggestion. We have removed the raw counts from the abstract and the main manuscript as requested and now focus the reporting on the more meaningful comparative data.

I suggest that the authors present the numbers of patients who were hospitalized due to infectious diseases and numbers for patients with HAIs. These numbers are more relevant than the absolute numbers of hospitalized patients. All outcomes for antimicrobial use should be reported in relation to the number of hospitalized patients with infectious diseases and HAIs. In the present form, all numbers are presented in relation to all hospitalized patients, which is not appropriate.

Response: Thank you for this suggestion. We have added the absolute numbers of patients with healthcare-associated infections (HAIs) to Table 3, as this data was systematically documented. However, data on patients hospitalized specifically for community-acquired infectious diseases were not available in our retrospective dataset. Our study is an observational cohort that includes all hospitalized patients, as any one of them could develop an HAI, which is the primary focus of our analysis. Therefore, we have maintained the total number of hospitalized patients as the denominator for antimicrobial use outcomes to reflect the true exposure and risk within the entire cohort.

The study would strongly benefit if the authors would include data from the post-pandemic (i.e. after 2022). It would be interesting to see whether the observed effects during the pandemic may have ‘normalized’ to pre-pandemic numbers. These results would actually be much more interesting than the current limited analysis. There are already dozens of studies reporting the effect of the pandemic on antimicrobial use and infectious disease prevalence.

Response: Thank you very much for your comment. We agree that including post-pandemic data (after 2022) would be highly valuable to observe any potential "normalization" of the trends we identified. The primary aim of our current study, however, was to provide a comprehensive, integrated analysis of the challenges to antimicrobial stewardship during the pandemic itself. We believe our results offer useful information for shaping institutional policies on antimicrobial stewardship programs, epidemiological surveillance, and financial planning for future health crises. We are planning to extend this analysis into the post-2022 period in a subsequent study to address the very important question of long-term recovery. We have added it as a limitation (see on page 14).

The discussion of the limitations is very weak. This study has multiple limitations that should be discussed, including their effects on the results. Some limitations are the single-centre design, the observational design, the retrospective design, and the lack of ‘before’ and ‘after’ data that allow a comparison of the patient populations.

Response: We have revised the limitation section, adding more details one page 14: This study has several limitations. The single center design limits generalizability and prevents the establishment of causal relationships. While the pandemic is a major explanatory factor for the observed changes, our study design cannot exclude the influence of other concurrent factors. These may include shifts in infection prevention and control measures, and adaptations in local treatment policies, all of which could have independently affected antimicrobial use and other outcomes. Furthermore, the absence of post-pandemic (post-2022) data means we cannot yet assess whether the identified trends have persisted or normalized. While we used standardized metrics, the retrospective nature of the data collection is subject to information bias. We attempted to mitigate analytical bias by using appropriate time-series analysis and checking regression model assumptions.”

The authors should revise their manuscript according to the STROBE guidelines (STrengthening the Reporting of OBservational studies in Epidemiology). Please provide the checklist with the revised manuscript.

Response: We have revised the manuscript in accordance with the STROBE (Strengthening the Reporting of Observational Studies in Epidemiology) guidelines and provided the completed checklist with the revised submission.

The caption of Table 3 is wrong.

Response: Thank you for pointing that out. We corrected it to: “Table 3. Healthcare-associated infections before and during the COVID-19 pandemic.”

It is unclear whether the hospital is a tertiary or quaternary referral hospital. The authors use both terms in their manuscript.

Response: Thank you for pointing that out. We have corrected it.

Round 2

Reviewer 1 Report

Comments and Suggestions for Authors

Well done

Author Response

REVIEWER 1

Open Review

Quality of English Language

(x) The English could be improved to more clearly express the research.
( ) The English is fine and does not require any improvement.

Yes

Can be improved

Must be improved

Not applicable

Does the introduction provide sufficient background and include all relevant references?

( )

( )

(x)

( )

Is the research design appropriate?

( )

(x)

( )

( )

Are the methods adequately described?

( )

(x)

( )

( )

Are the results clearly presented?

( )

(x)

( )

( )

Are the conclusions supported by the results?

( )

(x)

( )

( )

Are all figures and tables clear and well-presented?

( )

(x)

( )

( )

Comments and Suggestions for Authors

Well done

Response: We appreciate every comment and suggestion provided to us.

Reviewer 3 Report

Comments and Suggestions for Authors

Thank you for the revised manuscript.

"This study has several limitations. The single center design limits generalizability and prevents the establishment of causal relationships" This statement is not correct. The retrospective observational design prevents the establishment of casual relationships. The single center design prevents generalizability.

The authors changed the conclusion of the manuscript main text. I also advice changing the conclusion of the abstract.

I still believe that a major limitation of the study is that the outcomes (antimicrobial use, etc.) are calculated in relation to all hospitalized patients. The correct denominator would be "all hospitalized patients with infections". Since the authors cannot exclude that there were more patients with (bacterial) infections during the pandemic, conclusions on the impact of pandemic on antimicrobial use cannot be made. This should be discussed and mentioned in the abstract and discussion. 

Author Response

REVIEWER 3

Open Review

Quality of English Language

( ) The English could be improved to more clearly express the research.
(x) The English is fine and does not require any improvement.

Yes

Can be improved

Must be improved

Not applicable

Does the introduction provide sufficient background and include all relevant references?

(x)

( )

( )

( )

Is the research design appropriate?

( )

( )

(x)

( )

Are the methods adequately described?

( )

(x)

( )

( )

Are the results clearly presented?

(x)

( )

( )

( )

Are the conclusions supported by the results?

( )

( )

(x)

( )

Are all figures and tables clear and well-presented?

(x)

( )

( )

( )

Comments and Suggestions for Authors

Thank you for the revised manuscript.

Response: We appreciate your comments.

"This study has several limitations. The single center design limits generalizability and prevents the establishment of causal relationships" This statement is not correct. The retrospective observational design prevents the establishment of casual relationships. The single center design prevents generalizability.

Response: We sincerely thank the reviewer for this precise and correct clarification. We have amended the text in the Limitations section to accurately demonstrate that the single-center design limits generalizability, while the retrospective observational design prevents the establishment of causal relationships. We have added the following sentence to the Limitation section in the Discussion on page 14, lines [447-449]: Its single-center design limits the generalizability of the findings to other settings.”

The authors changed the conclusion of the manuscript main text. I also advice changing the conclusion of the abstract.

Response: We thank the reviewer for this suggestion. We have now revised the conclusion of the abstract (on page 1): “Conclusions: The pandemic intensified antimicrobial use, resistance, and costs. While limited by its-single-center design and the use of all inpatients as the denominator, these findings highlight the important need for reinforced antimicrobial stewardship to build healthcare system resilience.

I still believe that a major limitation of the study is that the outcomes (antimicrobial use, etc.) are calculated in relation to all hospitalized patients. The correct denominator would be "all hospitalized patients with infections". Since the authors cannot exclude that there were more patients with (bacterial) infections during the pandemic, conclusions on the impact of pandemic on antimicrobial use cannot be made. This should be discussed and mentioned in the abstract and discussion. 

Response: We appreciate your comments. We agree with the reviewer that this could be a limitation, and we are grateful for the opportunity to address it. We have now incorporated this point directly into the Limitations section. We explicitly state that using all hospitalized patients as the denominator is a limitation, that the correct denominator would be patients with infections, and that our inability to track this broader infection burden limits our conclusions regarding the specific drivers of antimicrobial consumption. We have added the following sentence to the Limitation section in the Discussion on page 14, lines [453-459]: Another limitation is the use of all hospitalized patients as the denominator for antimicrobial consumption. The correct denominator for assessing the appropriateness of antibiotic use would be patients with infections. Since we cannot ascertain whether the proportion of patients with bacterial infections changed during the pandemic, our ability to draw definitive conclusions about the drivers of antimicrobial consumption is limited. We could only track device-associated infections, which does not represent the total burden of infections in the hospital.”

We have also mentioned it in the abstract (on page 1): “While limited by its single-center design and the use of all inpatients as the denominator, these findings highlight the important need for reinforced antimicrobial stewardship to build healthcare system resilience”.

Round 3

Reviewer 3 Report

Comments and Suggestions for Authors

I am satisfied with the revision.